# UniAdapt: A Universal Adapter for Knowledge Calibration

## Abstract

Large Language Models (LLMs) require frequent updates to correct errors and keep pace with continuously evolving knowledge in a timely and effective manner. Recent research in *model editing* has highlighted the challenges in balancing generalization and locality, especially in the context of *lifelong model editing*. We discover that inserting knowledge directly into the model often causes conflicts and potentially disrupts other unrelated pre-trained knowledge. To address this problem, we introduce UniAdapt, a universal adapter for knowledge calibration. Inspired by the Mixture of Experts architecture and Retrieval-Augmented Generation, UniAdapt is designed with a vector-assisted router that is responsible for routing inputs to appropriate experts. The router maintains a vector store, including multiple shards, to construct routing vectors based on semantic similarity search results. UniAdapt is fully model-agnostic and designed for seamless plug-and-play integration. Experimental results show that UniAdapt outperforms existing lifelong model editors and achieves exceptional results in most metrics.

## 1 Introduction

Large Language Models (LLMs) have shown their outstanding abilities in understanding and generating texts, resulting in widespread deployment across various applications with significant social impacts Vaswani (2017); Radford et al. (2018). Although LLM is trained with up-to-date and highly accurate data, it still can make mistakes Huang et al. (2023), generating hallucinated responses. Furthermore, its world knowledge may quickly become out-dated. Due to computational cost, retraining or fine-tuning the model frequently is impractical. This demands a *model editor* that corrects the errors and keeps pace with continuously evolving knowledge in a timely and effective manner.

In recent years, *model editing* has emerged as a highly effective method for updating knowledge within LLMs. It aims to insert or update the responses for certain target queries, referred to as *edits*, while ensuring that responses on unrelated queries remain intact. For instance, ROME Meng et al. (2022a) locates and edits knowledge within LLMs. It treats a multi-layer perceptron (MLP) as a key-value store, where the key encodes a subject and the value encodes knowledge about that subject. ROME uses rank-one modification to insert key-value pairs into the MLP module directly. In contrast to ROME, MEND Mitchell et al. (2021) trains a meta-network to edit the target LLM. The meta-network learns to generate parameter updates that adapt the LLM to perform well on a set of tasks, without modifying the original model weights. SERAC Mitchell et al. (2022) takes a different approach. It utilizes an external cache to store edits. Rather than modifying the LLM itself, SERAC retrieves relevant information from the cache to augment the model's responses when necessary. This allows SERAC to flexibly incorporate new knowledge without directly altering the LLM's parameters. While these approaches can apply multiple edits sequentially, they often encounter challenges such as over-fitting and a tendency to forget previous edits quickly. In a harder setting, known as *lifelong model editing*, the editor is expected to insert thousands of edits effectively. Multiple approaches have been proposed. GRACE Hartvigsen et al. (2024) constructs a codebook that caches the edits, enabling longer sequences of edits than prior works. In contrast, WISE Wang et al. (2024) employs a dual parametric memory scheme that consists of a main memory for pre-trained knowledge and a side memory for edited knowledge. It further introduces an activation routing mechanism that determines which memory to access when given a query, thus optimizing the knowledge retrieval process. MEMoE Wang & Li (2024b) and LEMoE Wang & Li (2024a) introduce an adapter based on the Mixture of Experts (MoE) architecture. Their rout-

ing algorithm performs classification based on anchor embeddings, which overlooks the relation in a sentence and thus considers a dataset-specific approach. Despite the extensive effort, existing methods still suffer from either limited success in achieving generalizability (i.e., successfully introducing the new knowledge) or locality (i.e., successfully maintaining the model performance on unrelated knowledge).

To address the above-mentioned problem, we introduce UniAdapt, a universal adapter leveraging the MoE Shazeer et al. (2017); Fedus et al. (2022) architecture and Retrieval-Augmented Generation (RAG) Lewis et al. (2020); Sachan et al. (2021); Asai et al. (2023) for knowledge calibration. UniAdapt edits a model by adding an adapter to the selected MLP layer, *never changing the model's weights*. The adapter comprises a vector-assisted router and multiple parallel experts. The core idea is that the router is responsible for routing relevant queries to the corresponding experts. Additionally, if no suitable expert is found, the output of the selected layer remains unaltered to save resources. To achieve this, the vector-assisted router maintains multiple shards of a vector store, storing the sentence embeddings of newly introduced knowledge. When a query is received, the router constructs a routing vector where each element represents the highest semantic similarity score regarding each shard. This routing vector determines which experts are activated to handle the current query. The output of our adapter is combined with the original output to achieve precise calibration. Overall, UniAdapt is a fully model-agnostic, plug-and-play, and cost-effective lifelong model editor.

Our contributions are summarized as follows.

- We analyze and identify the weakness of the existing lifelong model editors relying on memory, highlighting opportunities for potential enhancements.
- We develop UniAdapt, a lifelong model editor that is designed to route queries to the most relevant experts based on semantic similarity. Our architecture is model-agnostic.
- Our experiments show that UniAdapt outperforms existing lifelong model editors by a substantial margin. UniAdapt possesses the ability to memorize and generalize effectively, making it a superior choice for lifelong learning tasks.

## 2 PRELIMINARIES

This section presents an overview of lifelong model editing and reviews state-of-the-art approaches that leverage memory to enhance the editing process. We also introduce common metrics used to evaluate the editor's performance.

### 2.1 LIFELONG MODEL EDITING

The lifelong model editing task Hartvigsen et al. (2024); Wang et al. (2024) involves making numerous updates to a pre-trained model over time, ensuring that it consistently refreshes its knowledge and stays aligned with the fast-changing information encountered in everyday life. This task modifies an initial base model $f_{\theta_0}$, parameterized by $\theta$ at the time step 0, using a dataset $D_{\text{edit}} = \{(\mathcal{X}_e, \mathcal{Y}_e) \mid (x_1, y_1), \cdots, (x_T, y_T)\}$. Formally, at the time step $T$, the model editor, denoted by ME, inserts the T-th edit into the model $f_{\theta_{T-1}}$ and produces an edited model $f_{\theta_T}$. Let $\mathcal{P}(\cdot)$ be a function that rephrases $x$ to a set of semantic equivalent inputs (we assume $x \in \mathcal{P}(x)$). The task of lifelong model editing is defined as follows:

$$f_{\theta_T} = \text{ME}(f_{\theta_{T-1}}, x_T, y_T) \text{ s.t. } f_{\theta_T}(x) = \begin{cases} y_e & \text{if } x \in \mathcal{P}(x_e) \wedge (x_e, y_e) \in D_{edit} \\ f_{\theta_0}(x) & \text{otherwise.} \end{cases} \tag{1}$$

The edited model $f_{\theta_T}$ should produce a desired output $y_e$ for each in-scope input $x \in \mathcal{P}(x_e)$ and $(x_e, y_e) \in D_{edit}$, while maintaining the original model's performance $f_{\theta_0}(x)$ on an irrelevant input $(x, y) \in D_{irr}$ where $D_{irr} = \{(x, y) \mid x \notin \mathcal{P}(x_e), \forall x_e \in \mathcal{X}_e\}$. It also preserves knowledge from past edits $(x_{<T}, y_{<T}) \in D_{\text{edit}}$. Additionally, the result of applying $f_{\theta_T}$ to $x$ and $\mathcal{P}(x)$ should be identical.

To measure the efficiency of a model editor, the edited model is subject to evaluation using the following metrics.

| Method | Memory | | Router | |
|---|---|---|---|---|
| | Parametric | Retrieval | Algorithm | Input |
| SERAC Mitchell et al. (2022) | ✔ | ✔ | Binary classifier | Sentence embedding |
| GRACE Hartvigsen et al. (2024) | ✘ | ✔ | Clustering | Activation score |
| WISE Wang et al. (2024) | ✔ | ✔ | Activation routing | Activation score |
| MEMoE Wang & Li (2024b) | ✔ | ✘ | Knowledge anchor | Anchor embedding |
| LEMoE Wang & Li (2024a) | ✔ | ✘ | Knowledge anchor | Anchor embedding |
| UniAdapt | ✔ | ✔ | Vector-assisted routing | Sentence embedding |

Table 1: Different routing strategies of recent methods. Parametric memory encodes knowledge within the model's parameters, whereas retrieval memory stores information in an external memory system for future access. Sentence embeddings preserve the semantic meaning of entire sentences, while activation scores represent the outputs from the activation layers of the neural network. Anchor embedding is formed by combining the embeddings of entities (such as subjects and objects) in a sentence with token embeddings through a concatenation operation.

**Reliability**: The edited model $f_{\theta_T}$ should generate the expected responses on intended edits:

$$\mathbb{E}_{(x_e, y_e) \in D_{\text{edit}}} \ \mathbb{1}\{\arg\max_y f_{\theta_T}(y \mid x_e) = y_e\} \tag{2}$$

**Locality**: The edited model $f_{\theta_T}$ should retain original responses on inputs that are irrelevant to intended edits:

$$\mathbb{E}_{(x, y) \in D_{\text{irr}}} \ \mathbb{1}\{\arg\max_y f_{\theta_T}(y \mid x) = f_{\theta_0}(y \mid x)\} \tag{3}$$

**Generality**: The model $f_{\theta_T}$ should generalize edits over other semantic equivalent inputs:

$$\mathbb{E}_{(x_e, y_e) \in D_{\text{edit}}} \ \mathbb{1}\{\arg\max_y f_{\theta_T}(y \mid x) = y_e\} \text{ s.t. } x \neq x_e \wedge x \in \mathcal{P}(x_e) \tag{4}$$

## 2.2 LIFELONG MODEL EDITING USING MEMORY

Multiple recent methods, shown in Table 1, incorporate *memories* and *routing mechanisms* to process inputs efficiently. The router is crucial in detecting and forwarding inputs to designated memories. If an input falls inside the scope of the existing edits, the router forwards it to the designated memory, which contains the new knowledge, thereby increasing reliability and generality. Conversely, inputs that fall outside of the edits are routed to the original model, maintaining locality. Due to the importance of the router Zhou et al. (2022); Dikkala et al. (2023), we prioritize optimizing routing mechanisms over memory enhancements. In the following, we discuss existing efforts on improving both routing inputs and routing algorithms and justify the design choices that we make for developing our method.

**Routing Input**. Recent research opts for *activation scores*, *sentence embeddings*, or *anchor embeddings* to construct the routing vectors. In our method, we rely on sentence embeddings over activation scores and anchor embeddings for the following reasons. First, the works Geva et al. (2020); Dai et al. (2021) discover that activation scores at a specific block capture various patterns (i.e., *shallow, semantic, or shallow + semantic*). They also suggest that lower blocks capture shallow patterns, while upper blocks capture semantic patterns. However, there is no definitive evidence that the activation scores at any specific layer can effectively capture the complete semantics of the input. Anchor embedding enhances the classification algorithm within the router. However, this approach is dataset-specific. When applied to factual knowledge, anchor embedding overlooks the full sentence context, focusing only on the subject and objects. This may lead to misclassification if the relation between the entities changes. In contrast, sentence embeddings are widely recognized for their ability to compute the semantic similarity of the inputs Reimers (2019); Gao et al. (2021); Cer et al. (2018); Feng et al. (2020). Second, sentence embeddings are model-agnostic, which means that they remain the same across different target models (i.e., the models that we aim to edit). On the other hand, activation scores and anchor embeddings are model-specific, varying across different target models. This potentially compromises the generalizability of methods that rely on them.

**Routing Algorithm**. In recent studies, research on the routing algorithms primarily focuses on searching for thresholds for separating relevant and irrelevant input. In the binary classification settings, SERAC defines a single threshold $\beta = 0.5$ for any pair of inputs. In multi-class classification

Figure 1: The architecture of UniAdapt inspired by MoE architecture. UniAdapt contains a router and multiple parallel feed-forward layers (a.k.a experts), denoted as $FFN_1, FFN_2, \cdots, FFN_k$. The router maintains a vector store containing multiple shards labeled $S_1, S_2, \cdots, S_k$. The matching colors of shards and experts indicate that each expert may hold knowledge relevant to queries associated with the shard. In the inference phase, the router computes a routing vector to selectively choose appropriate $FFNs$, ensuring precise calibration of the original MLP's output (more details in 3.2).

settings, the clustering algorithm in GRACE creates multiple pairs of thresholds (i.e., deferral radius $\epsilon$) and corresponding cluster centers (i.e., key $\mathbb{K}_i$). For an input $x$, WISE computes its routing activation indicator $\Delta_x$ and compares it with a fixed threshold $\epsilon$ to either forward it to the main memory or a side memory. Additionally, the choice of the side memory is determined by the value of $\Delta_x$. In our work, we generalize the routing algorithms as a sub-class of MoE where a router aims to forward inputs to relevant experts.

To achieve an effective lifelong model editor, we design a model-agnostic adapter that harnesses the strength of sentence embeddings and the MoE architecture. By employing sentence embeddings, the adapter can capture the semantic meaning of inputs effectively. The MoE architecture operates without altering the model's parameters, minimizing the potential conflicts with other unrelated pretrained knowledge and preserving the overall performance.

## 3 METHOD

In this section, we present the details of UniAdapt, a universal adapter based on the MoE architecture and a vector-assisted routing strategy, as illustrated in Figure 1. UniAdapt is appended immediately after a selected MLP layer to calibrate the output.

### 3.1 UNIADAPT ARCHITECTURE

The core idea of UniAdapt is to introduce several MoE-style experts to facilitate knowledge updates and learning, while keeping all the original parameters of LLM frozen to maintain its original behavior. Figure 1 introduces the forward pass of UniAdapt. UniAdapt consists of a router and multiple parallel experts. This module is appended to the original MLP to calibrate the original knowledge. The outputs of all experts are aggregated as a weighted sum to produce the final output. This choice aligns with recent experimental findings based on knowledge probing technologies, i.e., the MLP layers store knowledge Geva et al. (2020). Unlike traditional MoE, the router has a vector store for sentence embeddings. Given a token $x_i$ within the input sequence $x = \{x_i\}_{i=1}^L$, our adapter with $K$ experts computes a gate decision vector $\mathcal{G}$ that decides which expert to send the token $x_i$ to. This is defined as follows.

$$\mathcal{G} = \mathrm{H} \circ \mathrm{Top}_k(R(x)) \tag{5}$$

where $R(\cdot)$ defines a routing strategy (refer to details in 3.2). Note that the router makes the routing decision based on the whole sentence $x$. Consequently, all tokens $x_i$ within the sentence $x$ are directed to the same experts. The function $\mathrm{Top}_k(\cdot)$ keeps only the top-k values and sets all others to zero. The function $H$ is the Heaviside step function that outputs 1 for any non-negative input and 0 otherwise. Once the gate decision vector $\mathcal{G}$ is obtained, the corresponding output $h_i$ is generated through a weighted aggregation of each expert's computation on $x_i$, as follows:

$$h_i = \sum_{k=1}^K \mathcal{G}_k \cdot W_k \cdot x_i \tag{6}$$

where $W_k$ represents the linear projection weights of the $k$-th expert, and the gate decision $\mathcal{G}_k$ determines the contribution of the $k$-th expert to the output $h_i$. For efficiency, experts with $\mathcal{G}_k = 0$ do not require computation.

Overall, the forward pass of the UniAdapt layer, combined with the frozen original parameters $W_0$, can be expressed as:

$$h_i = \underbrace{W_0 \cdot x_i}_{\text{old knowledge}} + \lambda \underbrace{\sum_{k=1}^{K} \mathcal{G}_k \cdot W_k \cdot \overbrace{(W_0 \cdot x_i)}^{\text{old knowledge}}}_{\text{knowledge update}} \tag{7}$$

where $\lambda$ is a non-negative weighting coefficient used to balance the old knowledge and the knowledge update. The formula (7) shows that UniAdapt can minimize the knowledge update by setting $\lambda$ close to 0 to retain the original output.

## 3.2 Vector-Assisted Router

The core concept of UniAdapt is that the router has its own vector store to streamline the routing process. Our goal is to direct inputs that share similar knowledge with the edits to the appropriate experts, while inputs unrelated to any edits will bypass expert activation, leaving the output unchanged. To achieve this, we start with training a router to distinguish between related and unrelated inputs using our modified loss function. Once trained, the router's parameters are frozen. We fine-tune the adapter to incorporate edits using the default loss function of the model. In the following, we introduce the details of the router.

**Router Construction**. Similar to the existing approaches De Cao et al. (2021); Mitchell et al. (2021; 2022), our vector-assisted router is trained with a dataset. To decide whether an input $x$ is in $\mathcal{P}(x_e)$ of some edit $x_e$, we introduce a threshold $\epsilon$. If the similarity score $\Delta(x, x_e) \geq \epsilon$, $x$ is considered an in-scope input of $x_e$. Otherwise, $x$ is irrelevant to $x_e$. Thus, we want the similarity scores of in-scope edits to be larger than out-scope edits by a large margin.

$$\min\{\Delta(x_i, x_e)\} \gg \max\{\Delta(x_o, x_e)\}, \forall x_e \in \mathcal{X}_e, x_i \in \mathcal{P}(x_e), x_o \notin \mathcal{P}(x_e) \tag{8}$$

Note that when the number of edits increases, we observe that even though the edit $x$ is related to $x_e$ and not to $x_a$, there are numerous cases where $\Delta(x, x_e) < \Delta(x, x_a)$. Therefore, we want to distinguish between in-scope edits of multiple edits. That is,

$$\min\{\Delta(x_i, x_e)\} \gg \max\{\Delta(x_i, x_a)\}, \forall x_e \in \mathcal{X}_e, x_a \in \mathcal{X}_e \wedge x_a \neq x_e, x_i \in \mathcal{P}(x_e) \tag{9}$$

To achieve both objectives in (8) and (9), we design a loss that is inspired by the multiple negative ranking loss Henderson et al. (2017). For a single in-scope edit $x_e \in \mathcal{X}_e$, we form a batch of $K$ sentence pairs that contain a positive pair $(x_e, x_i)$ where $x_i \in \mathcal{P}(x_e) \wedge x_i \neq x_e$ and $K - 1$ negative pairs $(x_e, x_a)$ where $x_a \in \mathcal{X}_e \wedge x_a \neq x_e$. The training goal is to minimize the data's approximated mean negative log probability. For a single batch, the loss is:

$$\mathcal{L} = -\frac{1}{K} \sum_{i=1}^{K} \left[ \Delta(x_e, x_i) - \log \sum_{a=1}^{K-1} e^{\Delta(x_e, x_a)} \right] \tag{10}$$

The loss aims to maximize the distance between a positive pair and multiple negative pairs. Note that the objective in (8) is typically satisfied by most pre-trained sentence embedding frameworks Reimers (2019); Gao et al. (2021). Therefore, fine-tuning them with the loss function in (10) is sufficient to produce accurate similarity scores.

**Routing Strategy**. Similar to SERAC, we need a memory to store the edits to make semantic similarity queries. Unlike SERAC, we aim to store sentence embeddings (rather than the sentences themselves) in a vector store, both to reduce memory usage and to ensure compatibility with a wide range of frameworks Douze et al. (2024); Johnson et al. (2019). An example illustrating the functionality of the router is shown in Figure 2.

We have multiple experts to handle input queries. A router is used to distribute the input queries, and only a few experts are activated to enhance knowledge capacity Wang et al. (2024). To efficiently

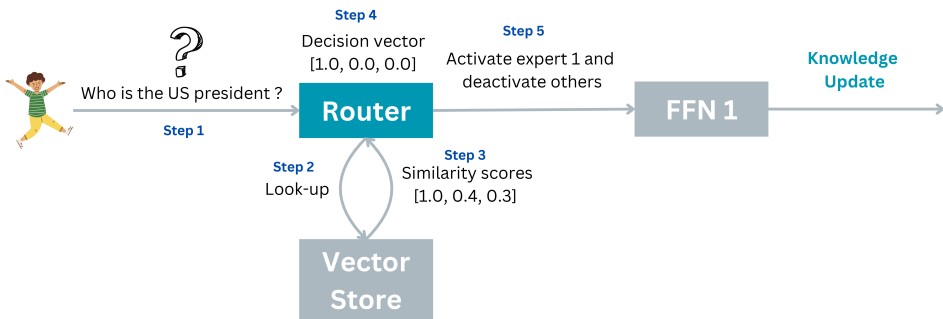

Figure 2: An example of the router's functionality, similar to a retriever in RAG. Instead of retrieving related documents, the router computes decision vectors based on the similarity scores. The similarity scores [1.0, 0.4, 0.3] indicate that there are three shards. The first shard has the highest similarity score thus the answer will be stored in expert 1 (also known as FFN1).

utilize these experts, we would like to dynamically route inputs to the most relevant experts and balance the number of edits calibrated by each expert. To achieve this goal, we propose a vector store sharding mechanism. We equally divide the embeddings of $N$ edits into $K$ shards, each shard stores around $N/K$ embeddings where $K$ is the number of experts. Given an input $x = \{x_i\}_{i=0}^{L}$ and a shard $S_k$, the router computes the routing score for each shard as follows:

$$\alpha_k = \max\{\Delta(x, x_e) \mid \forall x_e \in S_k\} - \epsilon \tag{11}$$

where $\epsilon$ is a non-negative threshold derived from the router construction step. The routing score is in the range $[-1, 1]$, if $\alpha_k$ is close to 1 then the input is the most similar to the shard $S_k$ and the router likely activates the expert $E_k$ to handle the input. If $\alpha_k \leq 0$ the expert $E_k$ is deactivated to reduce resource consumption. Given the routing scores for all shards, the decision vector is formed as follows:

$$R(x) = (\alpha_1, \dots, \alpha_j, \dots, \alpha_K) \tag{12}$$

## 4 EXPERIMENTS

In this section, we first present our experimental setup. Then, we discuss the performance of our method on two settings: *single editing* and *lifelong editing*.

### 4.1 EXPERIMENT SETUPS

**Datasets and Metrics**. We use two prominent model editing datasets: zsRE Levy et al. (2017) and Counterfact Meng et al. (2022a) for performance evaluation. zsRE is a context-free Question-Answering (QA) dataset built upon zero-shot relation extraction. Counterfact is a more challenging dataset containing factual knowledge with diverse subjects, relations, and linguistic variations. We evaluate the capability of UniAdapt using Reliability, Generality, and Locality (defined in Sect 2.1) along with the average scores over these metrics. Specifically, each edit record contains an editing pair $(x_e, y_e)$ along with a related edit $x_r$ and an unrelated edit $x_o$. The Reliability assesses if the edited model can recall the response $y_e$ from $x_e$. The Generality evaluates whether the edited model can produce $y_e$ given $x_r$. The Locality measures whether the edited model produces a consistent response for $x_r$ both before and after the edit.

**Baselines**. We compare UniAdapt with multiple recently proposed baselines. We categorize them into *non-memory based methods* including FT-L Meng et al. (2022a), MEND Mitchell et al. (2021), MEMIT Meng et al. (2022b) and *memory-based methods* including SERAC Mitchell et al. (2022), GRACE Hartvigsen et al. (2024), WISE Wang et al. (2024). Note that we exclude the results of MEMoe Wang & Li (2024b) and LEMoE Wang & Li (2024a), as their source code has not yet been made available.

| Method | Model | ZsRE | | | | Counterfact | | | |
|---|---|---|---|---|---|---|---|---|---|
| | | Reliability↑ | Generality↑ | Locality↑ | Score↑ | Reliability↑ | Generality↑ | Locality↑ | Score↑ |
| GRACE | | 0.34 | 0.00 | **1.00** | 0.45 | 0.00 | 0.00 | **1.00** | 0.33 |
| FT | | 0.57 | 0.30 | 0.88 | 0.58 | 0.93 | 0.16 | 0.73 | 0.61 |
| MEMIT | GPT2-XL | 0.65 | 0.50 | **1.00** | 0.72 | 0.62 | 0.24 | 0.99 | 0.62 |
| SERAC | | 0.43 | 0.29 | 0.85 | 0.52 | 0.44 | 0.01 | 0.95 | 0.47 |
| MEND | | 0.07 | 0.07 | 0.99 | 0.37 | 0.00 | 0.00 | 0.97 | 0.32 |
| **UniAdapt** | | **1.00** | **0.99** | **1.00** | **1.00** | **1.00** | **0.96** | 0.98 | **0.98** |
| GRACE | | 0.97 | 0.00 | 0.34 | 0.44 | **1.00** | 0.00 | 0.78 | 0.59 |
| FT | | 0.55 | 0.47 | 0.86 | 0.63 | 0.45 | 0.25 | 0.28 | 0.33 |
| SERAC | LLaMA2-7B | 0.52 | 0.41 | **1.00** | 0.64 | 0.45 | 0.12 | **1.00** | 0.52 |
| MEND | | 0.07 | 0.06 | 0.87 | 0.33 | 0.03 | 0.03 | 0.88 | 0.31 |
| WISE | | **1.00** | 0.94 | **1.00** | **0.98** | **1.00** | 0.76 | **1.00** | 0.92 |
| **UniAdapt** | | 0.97 | **0.96** | **1.00** | **0.98** | 0.97 | **0.95** | 0.98 | **0.97** |

Table 2: Main editing results with the number of edits $T = 1$. **Bold** is the best result, and underline is the second-best result.

FT-L is a direct fine-tuning method that aims to limit the extent of weight modifications. MEND is a meta-learning method that learns auxiliary models to predict weight changes in the editing model. MEMIT inserts thousands of key-value pairs into multiple layers of the network by considering a feed-forward layer as linear associative memory.

SERAC uses external memory to explicitly cache the edits and route an input query to either the counterfact model or the original model. GRACE replaces the hidden states of inputs if its activation scores fall inside a cluster of a codebook. WISE routes an input query to either side memories or the main memory using activation scores.

**Implementation Details**: We apply our edits to GPT2-XL and LLaMA2-7B. Our router is built on top of SBERT Reimers & Gurevych (2019) for similarity scores computation. We opt for two tasks: single editing and lifelong editing tasks. For single editing, following Meng et al. (2022a), the batch size is set to 5, we evaluate edits and roll back to the initial state after each batch of edits. For lifelong editing, the batch size is set to 5. We insert 1000 edits and evaluate without rolling back. For the baselines, WISE is only implemented for LLaMA2-7B and MEMIT is only implemented for GPT2-XL.

## 4.2 MAIN RESULTS

**Single Editing**. We evaluate the performance of UniAdapt in the single editing setting, $T$=1, and compute the average of 1000 runs. The evaluation results are shown in Table 2. We observe that Uni-Adapt consistently outperforms baselines across all tested models and most metrics. The results are balanced as it achieves scores of at least 0.97 in all metrics. In the zsRE setting, UniAdapt achieves scores of 1.00 and 0.98 on GPT2-XL and LLaMA2, respectively, achieving improvements of 28% and 0% over the second-best competitor. Similarly, the improvements are 36% and 5% in the Counterfact setting. A closer investigation shows that other tools often sacrifice their generality to achieve higher locality. GRACE and MEND achieve 0.0 in generality but 1.0 in the locality within the zsRE setting of GPT2-XL. Overall, this result demonstrates the efficacy and stability of UniAdapt's capability on handling a hard dataset (i.e., Counterfact).

Although the results of UniAdapt vary across different datasets like other baselines, it demonstrates consistent performance across different model architectures. Specifically, the difference remains below 3% in all metrics and under 2% in the average score. For the average score, the discrepancies in GRACE, FT, and SERAC range from 1% to 28%. FT is considered the least stable tool as its difference is 28%. In summary, the results indicate that UniAdapt not only achieves the highest scores but also maintains stability across diverse models.

**Lifelong Editing**. We evaluate the performance of UniAdapt in the lifelong editing setting, $T$=1000. The evaluation results are shown in table 3. The results clearly show a decline in the performance across all methods as $T$ increases from 1 to 1000. For example, FT and MEMIT experience a drop of over 50% and 20% respectively in almost all settings. This is attributed to the fact that new edits tend to overwrite previous ones. Among these methods, UniAdapt shows a negligible decline on the easier zsRE, and a significant advantage in terms of generalizing ability on Counterfact. A further

| Method | Model | ZsRE | | | | Counterfact | | | |
|--------|-------|------|------|------|------|------|------|------|------|
| | | Reliability↑ | Generality↑ | Locality↑ | Score↑ | Reliability↑ | Generality↑ | Locality↑ | Score↑ |
| GRACE | | 0.34 | 0.00 | **1.00** | 0.45 | 0.00 | 0.00 | **0.99** | 0.33 |
| FT | | 0.07 | 0.05 | 0.02 | 0.05 | 0.19 | 0.07 | 0.00 | 0.09 |
| MEMIT | GPT2-XL | 0.51 | 0.45 | 0.31 | 0.42 | 0.82 | **0.55** | 0.05 | 0.47 |
| SERAC | | 0.19 | 0.19 | 0.85 | 0.41 | 0.00 | 0.00 | 0.96 | 0.32 |
| MEND | | 0.21 | 0.20 | 0.99 | 0.47 | 0.00 | 0.00 | **0.99** | 0.33 |
| **UniAdapt** | | **0.98** | **0.93** | **1.00** | **0.97** | **0.98** | 0.53 | 0.91 | **0.81** |
| GRACE | | **0.98** | 0.01 | 0.34 | 0.44 | **0.99** | 0.00 | 0.77 | 0.59 |
| FT | | 0.16 | 0.14 | 0.04 | 0.11 | 0.04 | 0.01 | 0.01 | 0.02 |
| SERAC | LLaMA2-7B | 0.36 | 0.35 | **1.00** | 0.57 | 0.15 | 0.12 | **1.00** | 0.42 |
| MEND | | 0.29 | 0.29 | 0.85 | 0.48 | 0.15 | 0.12 | 0.96 | 0.41 |
| WISE | | 0.83 | 0.77 | **1.00** | 0.87 | 0.42 | 0.26 | 0.64 | 0.44 |
| **UniAdapt** | | 0.96 | **0.80** | **1.00** | **0.92** | **0.99** | **0.57** | 0.94 | **0.83** |

Table 3: Main editing results with the number of edits $T=1000$. **Bold** is the best result, and underline is the second-best result.

analysis reveals that UniAdapt significantly outperforms the nearest competitor by a large margin. In the GPT2-XL setting, UniAdapt has a remarkable gap of around 40% over MEMIT on the zsRE dataset. In the LLaMA2-7B setting, UniAdapt proves to be the best with around 40% difference compared to WISE in the Counterfact dataset. In both datasets, our overall score is the highest, significantly outperforming the other methods. Furthermore, while the lifelong editing setting has proved to be more challenging than the single editing setting, UniAdapt maintains impressive stability across models. The difference remains below 7% in all metrics and under 5% in the average score. In summary, UniAdapt excels at learning extensive new knowledge while preserving other unrelated pre-trained knowledge.

## 4.3 ABLATION STUDIES

In this section, we examine the effects of various hyper-parameters on the performance of UniAdapt. Given that zsRE has been extensively evaluated in numerous studies, we have implemented lifelong editing settings on the zsRE dataset with LLaMA2-7b.

**Effect of the Target Layer**. We conduct multiple experiments to *assess the impact of the choice of target layer on the performance*. We sequentially append UniAdapt to the MLP module of each transformer block and evaluate the performance of UniAdapt with 1000 edits. The results are illustrated in Figure 3a across various target layers. While locality remains stable, both reliability and generality encounter significant fluctuations, peaking at layer 3 and reaching their lowest point at the final layer. Our finding aligns with the work Zhao et al. (2024) that confirms the importance of editing the model at layer 3. Notably, regardless of the layer modified, generality consistently hits the lowest accuracy among all metrics, indicating that it is the most challenging metric to improve. Overall, performance tends to decline sharply as the target layer approaches the last layer.

**Effect of the Number of Experts**. We perform multiple experiments to study *how the number of experts impacts the performance*. Due to computational resource limitations, we sequentially set the number of experts to values in the range [1–10] and evaluate UniAdapt's performance with 1000 edits. Figure 3b illustrates the performance of UniAdapt with different numbers of experts. We find that the locality of model editing does not change with the number of experts, i.e., there is neither a decrease nor a performance improvement. This is expected because only relevant inputs are forwarded to experts. The reliability exhibits slight fluctuation (i.e., going upward and then downward) when the number of experts increases. Furthermore, it consistently remains above 0.95 across all scenarios. Unlike reliability and locality, the generalization of knowledge fluctuates with the number of experts, peaking when the number of experts is 4, i.e., increasing the number of experts initially boosts overall performance, but eventually leads to a decline. We hypothesize that the reason is that while having more experts can enhance recall by providing specialized knowledge, it may also make it more challenging for the router to effectively choose the most suitable experts.

**Effect of $\epsilon$**. We conduct multiple experiments to *evaluate the impacts of $\epsilon$ on the performance*. We sequentially set the $\epsilon$ to values in the range [0.1–0.9] and evaluate UniAdapt's performance after 1000 edits. Figure 3c depicts the performance of UniAdapt across various $\epsilon$. The results show that $\epsilon$ has little impact on the reliability and generality. In contrast, locality increases sharply as $\epsilon$ is raised

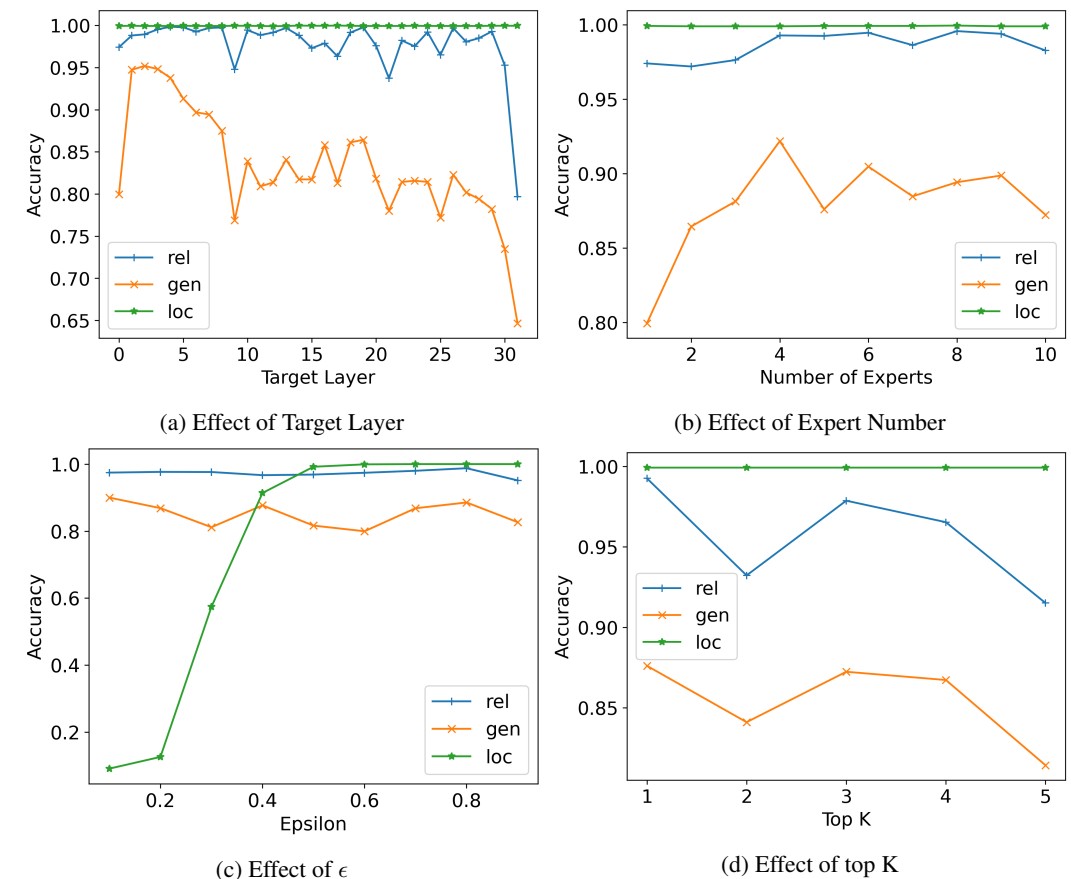

(a) Effect of Target Layer

(b) Effect of Expert Number

(c) Effect of $\epsilon$

(d) Effect of top K

Figure 3: The performances of UniAdapt regarding to different hyper-parameters where the notation *rel, gen, loc* are Reliability, Generality, and Locality respectively.

from 0.1 to 0.6. This can be attributed to the behavior of the router at low $\epsilon$ values. With a low $\epsilon$, the router tends to misclassify unrelated inputs, while relevant inputs remain unchanged. As $\epsilon$ increases, the router becomes more selective and only forwards inputs that are highly likely to be relevant, leading to higher locality.

**Effect of top-k routing**. We conduct multiple experiments to *evaluate the impacts of top-k routing on UniAdapt's performance*. We sequentially set $K$ to values in the range $[1–5]$, fix the number of experts at 5, and evaluate our performance after 1000 edits. Figure 3d depicts the performance of UniAdapt across various $K$. The results show that the locality remains unchanged across the different $K$ values. However, reliability and generality consistently decrease as $K$ increases. This suggests that while top-k routing does not impact locality, it hurts reliability and generality as the number of routing options increases. Interestingly, the best overall performance is achieved when $K$=1, indicating that using a single optimal routing path leads to the highest reliability and generality. As $K$ increases, the UniAdapt becomes less focused and may allocate resources to less relevant routing options, leading to decreased performance in terms of reliability and generality.

**Scale up to 6K**. We conduct multiple experiments to *assess the capability of UniAdapt on handling long continual edits*. We sequentially scale the number of edits to 2000, 3000, and 6000 and report our results along with WISE (the second-best competitor in our experiments) in Table 4. From the results, we observe that UniAdapt remains the best editor. WISE experiences a significant decline in both generality and reliability, dropping from 0.64 to 0.48 and 0.70 to 0.50 respectively. This is expected because WISE tends to incorrectly select the side memory when the number of edits increases. UniAdapt experiences a slight decrease of less than 0.02 in both metrics. Overall, the results highlight UniAdapt's exceptional performance on handling long continual edits, which makes it a practical solution.

| Method | T | Reliability↑ | Generality↑ | Locality↑ | Score↑ |
|--------|------|------|------|------|------|
| WISE | 2000 | 0.70 | 0.64 | **1.00** | 0.78 |
| UniAdapt | | **0.97** | **0.80** | 0.99 | **0.92** |
| WISE | 3000 | 0.64 | 0.58 | **1.00** | 0.74 |
| UniAdapt | | **0.96** | **0.77** | 0.99 | **0.91** |
| WISE | 6000 | 0.50 | 0.48 | **1.00** | 0.66 |
| UniAdapt | | **0.95** | **0.79** | 0.98 | **0.90** |

Table 4: Scaling to 6000 edits on zsRE dataset with LLaMA2-7b

## 5 RELATED WORK

Lifelong model editing is an active research area with many attempts Wang et al. (2024); Meng et al. (2022b); Yu et al. (2024) demonstrating encouraging results. Recently, MoE gained significant research attention for enhancing the performance of large language models. They have shown their potential in various applications. In the following, we highlight some of the most relevant works.

**Model editing**. UniAdapt is related to model editing which aims to update knowledge of pre-trained LLMs. Instead of retraining the model which is infeasible, the task of model editing is to fine-tune the model by either directly modifying the model parameters or dynamically loading new knowledge from external storage. MEND Mitchell et al. (2021) trains a meta-network that modifies the parameters of the target model. ROME Meng et al. (2022a) insert key-value pairs into a layer of a feed-forward layer by considering the layer as linear associative memory. While MEND and ROME are effective, they suffer from low locality. To address this, SERAC Mitchell et al. (2022) employs a router mechanism that directs inputs to the appropriate model (i.e., either the new model or the original model). IKE Zheng et al. (2023) teaches the targeted model to revise the output with high-quality demonstrations. Both SERAC and IKE achieve comparable results to MEND and ROME.

**Lifelong model editing**. UniAdapt is closely related to lifelong model editing, where thousands of edits are inserted continually. MEMIT Meng et al. (2022b) extends ROME to insert thousands of key-value pairs. GRACE Hartvigsen et al. (2024) assigns knowledge into multiple clusters, allowing the system to query and apply appropriate patches when needed. MELO Yu et al. (2024) extends GRACE by using dynamic Lora to store patches. WISE Wang et al. (2024) relies on activation scores to route inputs to either the main memory or side memory. Overall, these tools employ a routing mechanism, except for MEMIT. Both MEMoE Wang & Li (2024b) and LEMoE Wang & Li (2024a) rely on anchor embeddings to distribute tokens to the corresponding experts.

**Spare Mixture of Experts (SMoE)**. UniAdapt is closely related to SMoE, where a gate network or router is responsible for dispatching tokens to a subset of experts. The work Fedus et al. (2022) introduces an approach named *switch transformer* to scale neural networks up to a trillion parameters. It selectively activates relevant experts for each input. Shazeer et al. (2017) features a trainable gating network to optimize expert selection.

## 6 CONCLUSION

In this work, we present UniAdapt, a universal adapter for knowledge calibration. UniAdapt is fully model-agnostic and designed for seamless plug-and-play integration. It has MoE-style architecture and is attached to the MLP layer to calibrate the original output. The router with multiple shards can precisely forward queries to the experts that store knowledge and make no modifications when the queries are irrelevant. The experimental results show that UniAdapt achieves the significantly improved performance on various models and datasets.

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

# A  APPENDIX

## A.1  DESCRIPTION OF DATASETS

We utilized two standard datasets: zsRE Levy et al. (2017) and Counterfact Meng et al. (2022a). Table 5 illustrates examples from these datasets, where each row has three pairs: $(x_e, y_e)$, $(x_{irr}, y_{irr})$ and $(\mathcal{P}(x_e), y_e)$ for the evaluation. ZsRE is a context-free Question-answering (QA) dataset containing factual information. In contrast, Counterfact focuses on counterfactual information. Compared to zsRE, the Counterfact dataset is considered more challenging to apply, as it attempts to erase the model's existing contradictory information. Consequently, it often yields lower accuracy. In our experiments with these datasets, we adopt the version proposed by Yao et al. (2023)

| # | zsRE | Counterfact |
|---|---|---|
| $x_e, y_e$ | Which college or university is related with Mobolaji Johnson? **Royal Military Academy Sandhurst** | The native language of Francis Jammes is **German** |
| $x_{irr}, y_{irr}$ | nq question: where were the olympics held in the 1980s? **Moscow, Soviet Union** | The mother tongue of Frédéric Bastiat is **French** |
| $\mathcal{P}(x_e), y_e$ | Which university or university is associated with Mobolaji Johnson? **Royal Military Academy Sandhurst** | Where Francis Jammes is from, people speak the language of **German** |

Table 5: Editing dataset example

## A.2  TRAINING DETAILS

In our reported results in Table 2 and Table 3, UniAdaptis reported with the following hyper-parameters: number of experts = 1, $\epsilon = 0.6$, TopK = 1, edited layer = 0, and number of epochs to train the adapter = 25. It is worth noting that this configuration is not our best — our optimal setup uses an edited layer of 3 and 4 experts.

## A.3  ADDITIONAL EXPERIMENTS

In general, an adapter's effectiveness heavily depends on the layers selected for editing. Choosing the right layer for a specific dataset is crucial to achieving high accuracy. In addition to the results presented in the main content, we explored modifying different layers of two primary models: GPT2-XL and LLaMA2-7B, to identify the optimal layer for editing. Table 6 shows that for GPT2-XL, layer 16 achieves the highest score of 0.83, with layers 1 and 17 tying for second at 0.82. For LLaMA2-7B, layer 4 performs best, followed closely by layer 3. Overall, the best layer for editing varies between models. However, layer 0 emerges as a reliable choice, consistently yielding relatively high accuracy across models. Moreover, earlier layers typically yield better results than later ones.

| Layer | GPT2-XL | | | | LLaMA2-7B | | | |
|---|---|---|---|---|---|---|---|---|
| | Reliability↑ | Generality↑ | Locality↑ | Score↑ | Reliability↑ | Generality↑ | Locality↑ | Score↑ |
| 0 | 0.98 | 0.53 | **0.91** | 0.81 | 0.99 | 0.57 | 0.94 | 0.83 |
| 1 | **1.00** | 0.55 | **0.91** | 0.82 | **1.00** | 0.70 | 0.94 | 0.88 |
| 2 | **1.00** | 0.50 | **0.91** | 0.80 | **1.00** | 0.77 | 0.94 | 0.90 |
| 3 | **1.00** | 0.35 | **0.91** | 0.75 | **1.00** | 0.79 | 0.94 | 0.91 |
| 4 | **1.00** | 0.47 | **0.91** | 0.80 | 0.98 | **0.83** | 0.94 | **0.92** |
| 5 | **1.00** | 0.27 | **0.91** | 0.73 | 0.98 | 0.72 | 0.94 | 0.88 |
| 6 | 0.82 | 0.24 | **0.91** | 0.66 | 0.99 | 0.68 | 0.94 | 0.87 |
| 7 | **1.00** | 0.41 | **0.91** | 0.77 | 0.96 | 0.65 | 0.94 | 0.85 |
| 8 | **1.00** | 0.47 | **0.91** | 0.79 | 0.99 | 0.62 | 0.94 | 0.85 |
| 9 | **1.00** | 0.52 | **0.91** | 0.81 | 0.99 | 0.56 | 0.94 | 0.83 |
| 10 | **1.00** | 0.51 | **0.91** | 0.81 | 0.88 | 0.33 | 0.94 | 0.72 |
| 11 | **1.00** | 0.53 | **0.91** | 0.81 | 0.98 | 0.47 | 0.94 | 0.80 |
| 12 | **1.00** | 0.46 | **0.91** | 0.79 | 0.98 | 0.51 | 0.94 | 0.81 |
| 13 | **1.00** | 0.43 | **0.91** | 0.78 | 0.94 | 0.43 | 0.94 | 0.77 |
| 14 | 0.94 | 0.42 | **0.91** | 0.76 | 0.99 | 0.45 | 0.94 | 0.79 |
| 15 | **1.00** | 0.42 | **0.91** | 0.78 | 0.95 | 0.35 | 0.94 | 0.75 |
| 16 | **1.00** | **0.57** | **0.91** | **0.83** | 0.99 | 0.49 | **0.95** | 0.81 |
| 17 | **1.00** | 0.55 | **0.91** | 0.82 | 0.93 | 0.38 | 0.94 | 0.75 |
| 18 | **1.00** | 0.37 | **0.91** | 0.76 | 0.99 | 0.45 | 0.94 | 0.80 |
| 19 | **1.00** | 0.53 | **0.91** | 0.81 | 0.96 | 0.41 | 0.94 | 0.77 |
| 20 | **1.00** | 0.39 | **0.91** | 0.77 | 0.99 | 0.47 | 0.94 | 0.80 |
| 21 | **1.00** | 0.33 | **0.91** | 0.75 | 0.97 | 0.42 | 0.94 | 0.78 |
| 22 | **1.00** | 0.53 | **0.91** | 0.81 | 0.98 | 0.42 | 0.94 | 0.78 |
| 23 | **1.00** | 0.40 | **0.91** | 0.77 | 0.99 | 0.46 | 0.94 | 0.80 |
| 24 | **1.00** | 0.53 | **0.91** | 0.81 | 0.99 | 0.47 | 0.94 | 0.80 |
| 25 | **1.00** | 0.36 | **0.91** | 0.76 | 0.96 | 0.42 | 0.94 | 0.78 |
| 26 | **1.00** | 0.48 | **0.91** | 0.80 | 0.97 | 0.42 | 0.94 | 0.78 |
| 27 | **1.00** | 0.46 | **0.91** | 0.79 | 0.96 | 0.39 | 0.94 | 0.76 |
| 28 | 0.98 | 0.45 | **0.91** | 0.78 | 0.88 | 0.32 | 0.94 | 0.72 |
| 29 | 0.53 | 0.16 | **0.91** | 0.54 | 0.99 | 0.42 | 0.94 | 0.78 |
| 30 | 0.99 | 0.40 | **0.91** | 0.77 | 0.87 | 0.32 | 0.94 | 0.71 |
| 31 | **1.00** | 0.47 | **0.91** | 0.80 | 0.70 | 0.30 | 0.94 | 0.65 |
| 32 | **1.00** | 0.33 | **0.91** | 0.75 | | | | |
| 33 | **1.00** | 0.29 | **0.91** | 0.73 | | | | |
| 34 | **1.00** | 0.30 | **0.91** | 0.74 | | | | |
| 35 | 0.99 | 0.26 | **0.91** | 0.72 | | | | |
| 36 | 0.97 | 0.28 | **0.91** | 0.72 | | | | |
| 37 | 0.98 | 0.28 | **0.91** | 0.72 | | | | |
| 38 | 0.99 | 0.26 | **0.91** | 0.72 | | | | |
| 39 | 0.91 | 0.20 | **0.91** | 0.68 | | | | |
| 40 | 0.95 | 0.25 | **0.91** | 0.70 | | | | |
| 41 | 0.92 | 0.22 | **0.91** | 0.68 | | | | |
| 42 | 0.94 | 0.21 | **0.91** | 0.69 | | | | |
| 43 | 0.93 | 0.21 | **0.91** | 0.69 | | | | |
| 44 | 0.89 | 0.20 | **0.91** | 0.67 | | | | |
| 45 | 0.91 | 0.22 | **0.91** | 0.68 | | | | |
| 46 | 0.93 | 0.21 | **0.91** | 0.68 | | | | |
| 47 | 0.82 | 0.17 | **0.91** | 0.63 | | | | |

Table 6: Counterfact dataset. Editing performance across all layers