# OpenReview forum: "UniAdapt: A Universal Adapter for Knowledge Calibration"
_ICLR.cc/2025/Conference — Submitted to ICLR 2025_

### Official Review · Reviewer_p1SB · 2024-10-31

**Soundness:** 3
**Presentation:** 2
**Contribution:** 2
**Rating:** 6
**Confidence:** 4

**Summary:**

The paper addresses the challenge of updating and maintaining Large Language Models (LLMs) in a timely and effective manner.  To mitigate the lifelong editing problem, they introduce UniAdapt, a universal adapter designed for knowledge calibration. UniAdapt leverages a Mixture of Experts (MoE) architecture and Retrieval-Augmented Generation (RAG) to route inputs to the most relevant experts based on semantic similarity. The router maintains a vector store with multiple shards to construct routing vectors, ensuring precise calibration of the model's output. The adapter is designed to be fully model-agnostic and plug-and-play, making it versatile for various LLMs. Experimental results demonstrate that UniAdapt outperforms existing lifelong model editors in most metrics, showcasing its effectiveness in balancing generalization and locality.

**Strengths:**

1. The experimental results are compelling, showing that UniAdapt outperforms existing lifelong model editors. The paper provides detailed metrics and comparisons, demonstrating the adapter's ability to memorize and generalize effectively.

**Weaknesses:**

- The citation format in the paper is not correct, making the reading difficult.
- The baseline is not comprehensive, the author just compared it with the WISE but other MoE methods like MEMoE and LEMoE are not considered. I think this would weaken the discussion in Table 1 and the contribution of the work as MoE is not a novel thing and the contribution should be more clear.
- The vector store here would be extra parameters and when the number of edits grows this would lead to more memory requirement and time computing, which makes me concerned about the efficiency of the proposed method. It's better to show an Inference Time Analysis.

**Questions:**

- When scaling up to 6k, what's the MoE setting such as the TopK and number of the experts.

---

> ### Author Response · Authors · 2024-11-25
>
> We appreciate the reviewer's valuable comments and suggestions, which will help us improve our work.
>
> > **Q1: citation format**
>
> Thank you for pointing this out. We have double-checked many other papers, and they appear to use the same citation format as we do. We will review this again and make any necessary modifications in the next version.
>
> > **Q2: missing MEMoE and LEMoE**
>
> While we wanted to compare with these models directly, their source code was not publicly available at the time of our experiments. Nevertheless, their reported results under the same settings (ZsRE, Llama7b, 1000 edits) were significantly lower than ours:
>
> | Model      | Rel  | Gen  | Loc  | Avg  |
> |------------|------|------|------|------|
> | MEMoE      | 0.70 | 0.43 | **1.00** | 0.71 |
> | LEMoE      | 0.80 | 0.60 | **1.00** | 0.82 |
> | UniAdapt   | **0.96** | **0.80** | **1.00** | **0.92** |
>
> > ** Q3: resource consumption**
>
> For detailed information about inference time, training time, and memory usage, please refer to our response to all reviewers. Our experimental results demonstrate the efficiency of our approach.
>
> > **Q4: 6K edits configuration**
>
> To maintain consistency with our experiments in Tables 2 and 3, we used the same settings: topK = 1 and the number of experts = 1. As shown in our ablation results, our performance could potentially improve by increasing these values to topK = 4 and experts = 4.

---

> ### Comment · Reviewer_p1SB · 2024-11-25
> **Response to the rebuttal**
>
> Thanks for your response.
>
> It solve my concerns and I decide to raise my score.

---

> > ### Author Response · Authors · 2024-11-26
> >
> > Thank you! We are encouraged by your acknowledgment.

---

### Official Review · Reviewer_Qpvw · 2024-11-01

**Soundness:** 3
**Presentation:** 3
**Contribution:** 2
**Rating:** 5
**Confidence:** 5

**Summary:**

This paper proposes  a universal adapter for knowledge calibration (UniAdapt). The experimental results show that UniAdapt achieves the significantly improved performance on various models and datasets.

**Strengths:**

UniAdapt is fully model-agnostic and designed for seamless plug-and-play integration.

**Weaknesses:**

The experimental analysis is not sufficiently thorough:

 - There is no analysis of the resource consumption of different methods, such as inference time and memory usage.
 -  The baselines are inconsistent across different vanilla models. Specifically, in Table 2, different base models use different baselines—for example, WISE is only applied to LLaMA2-7B, and MEMIT only to GPT2-XL. Why did the author choose this experimental setup?
 -  There is a lack of Out-of-Distribution evaluation.

**Questions:**

The author states that the proposed method primarily aims to address the conflicts caused by directly inserting knowledge into the model. Which experimental result in the paper demonstrates the mitigation of this issue?

---

> ### Author Response · Authors · 2024-11-25
>
> We appreciate the reviewer's valuable comments and suggestions, which will help us improve our work.
> > **Q1: resource consumption**
>
> For detailed information about inference time, training time, and memory usage, please refer to our response to all reviewers. Our experimental results demonstrate the efficiency of our approach.
>
> > **Q2: missing baselines**
>
> Thank you for pointing out these missing in our paper. While we wanted to include these comparisons in our table, several technical constraints made this infeasible. First, MEMIT lacks an official implementation for Llama7b. The unofficial version requires an expensive co-invariant computation process for the Wikipedia dataset on the edited layer—our attempts to compute this ran for two days before hanging. For GPT2-XL, we used the official version (approximately 7.5GB) provided by the authors. Second, WISE is only officially implemented for LLama2-7b. Though we contacted the authors for a GPT2-XL implementation, they provided one but expressed uncertainty about its reliability. This uncertainty led us to omit WISE results for GPT2-XL. Based on our analysis, if the implementation is accurate, WISE would likely rank as the second-best tool for GPT2-XL.
>
> > **Q3: out of distribution**
>
> Thank you for your suggestions. Here are our out-of-distribution evaluation results based on the setup and dataset (75 records) provided by the WISE paper.
>
> | Model      | Rel  | ood  | Loc  |
> |------------|------|------|------|
> | W/o edit   | 0.69 | 0.38 | **1.0** |
> | FT         | 0.53 | 0.05 | 0.53 |
> | Grace      | 0.96 | 0.41 | 0.07 |
> | WISE       | **0.98** | **0.53** | **1.0**  |
> | UniAdapt   | **0.98** | 0.41 | **1.0**  |
>
> It's not surprising that our implementation cannot outperform WISE. Our main focus was the routing algorithm, and with simple routing input, our implementation essentially functions as a LoRA adapter. The provided dataset favors WISE's implementation since WISE excels at memory storing rather than routing. In the dataset, routing inputs are short (around 5 words) and completely distinct from each other, allowing WISE's routing component to easily achieve 100% accuracy while focusing solely on storage. We acknowledge that our memory-storing component has limitations and considerable room for improvement.
>
> > **Q4: addressing the knowledge conflicts**
>
> Rather than directly addressing knowledge insertion conflicts in the model, we take another approach by introducing additional external memories (more experts) to store new knowledge instead of modifying the network parameters directly. As shown in Figure 3B, increasing the number of experts to 4 achieves the highest accuracy across rel, gen, and loc metrics. This suggests that adding more memories can help reduce potential conflicts.

---

> > ### Comment · Reviewer_Qpvw · 2024-11-27
> > **keep  original score**
> >
> > Thank you for your response. I’ve decided to keep my original score.

---

> > > ### Author Response · Authors · 2024-11-27
> > >
> > > We appreciate your valuable comments and suggestions

---

### Official Review · Reviewer_8c67 · 2024-11-01

**Soundness:** 3
**Presentation:** 3
**Contribution:** 2
**Rating:** 5
**Confidence:** 3

**Summary:**

The paper presents a novel approach to updating LLMs with new information without compromising existing knowledge. Traditional model-editing techniques often disrupt previous knowledge or fail to generalize well. UniAdapt addresses this by introducing a MoE architecture combined with a vector-assisted router, which selectively directs relevant queries to specific ``experts'' based on semantic similarity. This method allows seamless updates without altering the model’s core parameters, preserving unrelated knowledge effectively.

Experimental results on datasets such as zsRE and Counterfact demonstrate UniAdapt’s reliability in handling updates, generalizing edits, and maintaining unrelated knowledge compared to other methods. By keeping model stability high across thousands of sequential edits, UniAdapt emerges as a scalable, model-agnostic solution for maintaining LLMs with up-to-date knowledge, ideal for lifelong learning in dynamic environments.

**Strengths:**

1. UniAdapt's plug-and-play nature allows it to integrate with various LLMs without altering their original parameters, making it an adaptable and versatile solution for different base models.

2. By combining MoE with a vector-assisted router, UniAdapt selectively routes updates to specific ``experts'', effectively preserving existing knowledge while incorporating new information. This selective routing maintains a balance between reliability and generality.

3. The paper provides experimental evidence to show that UniAdapt performs better than baseline methods. UniAdapt performs well in datasets like zsRE and Counterfact.

**Weaknesses:**

1. The experiments were limited to GPT2-XL and LLaMA2-7B, which may not be sufficient to generalize the results. It would be helpful to include results on state-of-the-art LLMs across a range of model sizes (e.g., 13B and 70B) for more comprehensive insights.

2. Regarding the effect of the target layer, it would be interesting to explore the effects of editing multiple layers simultaneously. Since layer behaviors might differ across models, further investigation on other models would be valuable to confirm similar patterns. Without this, the generalizability of the findings may be constrained.

3. UniAdapt's vector-assisted routing adds complexity, which may increase computational overhead. It would be beneficial to provide cost analysis for both training and inference.

**Questions:**

See above.

---

> ### Author Response · Authors · 2024-11-25
>
> We appreciate the reviewer's valuable comments and suggestions, which will help us improve our work.
> > **Q1: experiments on large models (13B, 70B)**
>
> Thank you for your suggestions to make our experiments more comprehensive. While our current experiments align with previous work, we face several constraints. Due to time limitations during the rebuttal phase and resource constraints, we cannot run experiments on bigger models. Additionally, we lack implementations of baselines for large models, which prevents precise comparisons. We hope to include these additional experiments in future versions of the paper.
>
> > **Q2: edit multiple layers**
>
> We thank you for your suggestions to make our experiments more comprehensive. According to our knowledge, there are only a limited number of papers that aim to modify multiple layers, such as MEMIT. We will explore your suggestions in our future work, as we believe that editing multiple layers may improve the ability to store new knowledge.
>
> > **Q3: routing complexity**
>
> Our router training introduces independence rather than complexity. By training the router once and applying it to multiple models and testing configurations, we significantly reduce training time. Our router training requires less than 1 minute for 1,000 edits and 3 minutes for 3,000 edits. For detailed information about inference time, training time, and memory usage, please refer to our response to all reviewers. Our experimental results demonstrate the efficiency of our approach.

---

> > ### Comment · Reviewer_8c67 · 2024-11-26
> > **Response to the authors**
> >
> > Thank you for your responses. I am satisfied with the response regarding the routing complexity. However, I still believe more experimental results and variations could make the study more comprehensive.

---

> > > ### Author Response · Authors · 2024-11-27
> > >
> > > Thank you for your valuable feedback! I'm pleased that the routing complexity was clearly explained. We will conduct further experiments to strengthen our research findings.

---

### Official Review · Reviewer_RvYS · 2024-11-02

**Soundness:** 3
**Presentation:** 3
**Contribution:** 2
**Rating:** 6
**Confidence:** 3

**Summary:**

The main contribution of the paper is the development of UniAdapt, a universal adapter for lifelong model editing in Large Language Models (LLMs) that efficiently integrates new knowledge. The pursuit of lifelong model editing for LLMs is highly pertinent given the growing demand for adaptable AI systems. The concept is crucial for developing models that can continually learn and adapt without requiring full retraining.UniAdapt uses semantic similarity to route queries to the appropriate experts, significantly improving the model’s ability to adapt and generalize over traditional methods.

**Strengths:**

1. Lifelong model editing for LLMs is a promising research direction. The concept is crucial for developing LLM that can continually learn and adapt without requiring full retraining.

2. The experimental results demonstrate the effectiveness of the UniAdapt framework. By integrating a Mixture of Experts with the idea of  RAG, the paper shows improvements in model adaptability and performance on benchmarks designed to test these aspects.

**Weaknesses:**

**Weakness 1**

The core motivation, as stated in the abstract:
>“We discover that inserting knowledge directly into the model often causes conflicts and potentially disrupts other unrelated pre-trained knowledge”

This motivation lacks novelty as reducing the disruption to pre-trained knowledge while inserting new information is a foundational objective of the entire knowledge editing field (*measured by the locality metric*), as extensively discussed in foundational papers such as “Fast Model Editing at Scale.”



**Weakness 2**

The paper’s primary contribution is applying the concept of a Mixture of Experts to the knowledge editing field. While technically sound, this application does not introduce many new insights into the field. Moreover, the implementation is described as complex and the overall concept may not engage the community due to its incremental nature rather than a groundbreaking innovation.


While the paper presents valid improvements in certain performance metrics for the chosen datasets, the novelty is somewhat limited, and the implementation is not elegant. Considering the high standards of ICLR, the paper could be seen as a borderline case. My overall score would be 5.5, reflecting a neutral to slightly positive evaluation but acknowledging the concerns regarding novelty and implementation complexity.

**Questions:**

Please refer to the previous section.

---

> ### Author Response · Authors · 2024-11-25
>
> We appreciate the reviewer's valuable comments and suggestions, which will help us improve our work.
> > **Weakness 1: Motivation discussion**
>
> Our sentence indicates that inserting new knowledge may disrupt unrelated pre-trained knowledge - a well-known challenge in the field. It is consistent with the motivation to reduce the disruption to pre-trained knowledge while inserting new information. For this reason, we designed a UniAdapt architecture that uses a router to activate knowledge updates through a precise routing algorithm.
>
> > **Weakness 2: Contribution justification**
>
> Thank you for your comments about our approach. While our paper may not present a ground-breaking idea, it makes significant novel contributions beyond incremental research. To the best of our knowledge, only three papers—WISE, MEMoE, and LEMoE—are closely related to our work. We advance the field by generalizing the architecture of routers and memories while clearly identifying the differences between these previous approaches. Our analysis revealed two key areas for improving overall performance: the routing algorithm and the method of storing data in external memories. Our paper focuses primarily on the routing algorithm aspect. Our main contribution is an independent router that can be trained once and attached to any model for classification (as demonstrated with both GPT2 and LLama7b)—a significant improvement over WISE's approach, which requires retraining the router for different models. We also contribute a novel loss function that enhances routing performance. Our experiments show impressive routing results, achieving 0.999 accuracy on the ZsRE dataset and 0.985 on the Counterfact dataset with a threshold of 0.6 and 3K edits.

---

> > ### Comment · Reviewer_RvYS · 2024-11-25
> >
> > Thanks for the reply!
> >
> > I think my rating and score are both fair and supportive. Best of luck with your submission!

---

> > > ### Author Response · Authors · 2024-11-26
> > >
> > > Thank you for the support and kind words!

---

### Author Response · Authors · 2024-11-25

We thank all reviewers for their valuable comments and suggestions that will help improve our work. We are encouraged that the reviewers found our method novel and well-demonstrated, with comprehensive and robust experiments.

We address their concerns as follows:

- **Contribution**: We generalize the problem and propose an independent router that can be trained once and applied to any model. Additionally, we introduce a loss function to enhance classification performance.
- **Resource consumption**: Our additional experiments demonstrate reasonable resource usage, with better training and inference times compared to WISE.

As all reviewers expressed interest in resource consumption experiments, we present them here to address their questions.

### **1. Inference time analysis**

We measured LLAMA2-7b's inference time with and without UniAdapt after training with T=3000 on ZsRE. Based on an average of three inference trials, the base model took 0.014 seconds. UniAdapt added a minor overhead of 5.75%—slightly higher than WISE-Merge (3%) but lower than WISE-Retrieve (7%).


### **2. Memory analysis**

UniAdapt loads two modules: a router built on top of all-MiniLM-L6-v2 and a vector storage for embeddings. The router requires 620 MB, while the original LLAMA2-7b model requires 26,222 MB. Each embedding has a shape of 384. For 3,000 embeddings of float32, the size is 3,000 × 384 × 4 = 4.6 MB. An expert requires 64 MB. With a single expert, the total additional memory needed is 688.6 MB, representing a 2.63% overhead. When scaling UniAdapt to 8 experts and 9,000 edits, the required memory becomes 4.6 × 3 + 620 + 64 × 8 = 1,145.8 MB, with a 4.37% overhead. The WISE’s overhead is 0.64% in theory and 4% in practice.

### **3. Training time analysis**

Due to time limitations, we only analyzed the training time of UniAdapt compared to WISE on the ZsRE dataset using LLaMA-7B.
| Method    | Number of edits | Router training (s) | Edit training (s) | Total (s) |
|-----------|------------------|----------------------|--------------------|-----------|
| UniAdapt  | 10              | 0.96                | 14.9              | **15.86**     |
| UniAdapt  | 100             | 6.08                | 142.8             | **148.88**    |
| UniAdapt  | 1000            | 55.35               | 1423.82           | **1479.17**   |
| WISE      | 10              | 0                   | 94                | 94        |
| WISE      | 100             | 0                   | 603.12            | 603.12    |
| WISE      | 1000            | 0                   | 5273.82           | 5273.82   |

UniAdapt's training time consists of two components: router training and edit training. While the training time increases with the number of edits, and UniAdapt requires additional time for router training, its total training time is still approximately 4.5 times faster than WISE.

---

### Meta-Review · Area_Chair_pufB · 2024-12-20

**Metareview:**

The main contribution of this paper is the development of UniAdapt, a universal adapter for lifelong model editing in Large Language Models (LLMs) that efficiently integrates new knowledge. The concept of lifelong model editing is highly relevant given the increasing demand for adaptable AI systems. This approach is crucial for developing models that can continuously learn and adapt without requiring full retraining. UniAdapt leverages semantic similarity to route queries to the appropriate experts, significantly enhancing the model's ability to adapt and generalize compared to traditional methods. However, the core motivation about reducing conflicts when inserting new knowledge into the model is not novel, as it has already been a key objective in the knowledge editing field, particularly addressed by foundational work like "Fast Model Editing at Scale." The application of the Mixture of Experts concept to knowledge editing is new but does not introduce  new insights. The contribution feels incremental, and the implementation is described as complex, potentially limiting its appeal to the community. It is recommended that the author revise the paper according to the reviewers' suggestions and include additional experiments.

**Additional Comments On Reviewer Discussion:**

The reviewer and the author had a discussion, and some reviewers think that the paper lacks innovation.

---

### Decision · Program_Chairs · 2025-01-22

Reject